# What Findings on Chest CTs Can Delay Diagnosis of Pleuropulmonary Paragonimiasis?

**Kai Ke Li [1], Gong Yong Jin [1,2,\*] and Keun Sang Kwon [3]**

[1] Department of Radiology, Jeonbuk National University Medical School, Jeonju 54896, Korea; lee616916836@gmail.com
[2] Research Institute of Clinical Medicine, Biomedical Research Institute of Jeonbuk National University Hospital, Jeonbuk National University Medical School, Institute of Medical Science, Jeonju 54970, Korea
[3] Department of Province, Jeonbuk National University Medical School, Jeonju 54896, Korea; drkeunsang@jbnu.ac.kr
\* Correspondence: gyjin@jbnu.ac.kr; Tel.: +82-063-250-2307

**Abstract:** Purpose: The purpose of this study was to investigate which findings were delayed in diagnosis with respect to chest CT findings of paragonimiasis. Methods: This retrospective, informed questionnaire study was conducted to evaluate chest CT scans of 103 patients (58 men and 45 women; mean age 46.1 ± 14.6 years). The patients were diagnosed with paragonimiasis from 2003 to 2008 in four tertiary hospitals. Statistical analysis was performed using the chi-square test to identify differences between an initially correct diagnosis and an incorrect one of paragonimiasis on chest CT scans, for which we evaluated such variables as the location of lesion, type of parenchymal lesions, and worm migration track. Results: Nodular opacities on chest CT scans were the most common findings (53/94, 56.4%). The sign of worm migration tracks was only present in 18.1% of cases (17/94). Although statistically insignificant, the form of consolidation (18/25, 72%) and mass (6/8, 75%) on CT was common in correct diagnostics, and the form of the worm migration track (12/17, 70.6%) was high in correct diagnostics. Conclusion: A delayed diagnosis of paragonimiasis may often be made in patients with non-nodular, parenchymal lesions who are negative for worm migration track on chest CT scans.

**Keywords:** parasitic disease; paragonimiasis; lung; computed tomography (CT)





## 1. Introduction

Pleuropulmonary paragonimiasis is a food-borne parasitic disease caused by trematodes belonging to *Paragonimus* spp., which is endemic to Asian, African, and South American countries such as China, Japan, Liberia, Nigeria, and Viet Nam [1]. This is a typically foodborne zoonotic helminthiasis. Human infections occur through consuming improperly cooked freshwater crustaceans, mostly crabs or crayfish, or eating paratenic hosts. According to the World Health Organization in 2015, approximately one million people are infected every year. Pleuropulmonary paragonimiasis has always been a serious medical problem in some countries, such as China, Japan, and Korea [2]. In South Korea, where pleuropulmonary paragonimiasis is endemic, the incidence of the disease has decreased recently as a result of improvements in public health and changes in dietary habits [3]. However, the disease has remained a potential threat and has become a neglected disease, as more than 10% of freshwater crabs sold in markets in Seoul were infected with the metacercariae of the parasite in the early 1990s [4], and even 89.8–96.5% of crayfish in Wando-gun and Haenam-gun were infected with P. *westermani* metacercariae [5]. Clinical cases of pleuropulmonary paragonimiasis are still reported [6,7].

In previous studies, the well-known computed tomography (CT) findings of pleuropulmonary paragonimiasis include pleural abnormalities, such as pleural effusion and

hydropneumothorax, and pulmonary abnormalities, such as air-space consolidation, worm-migration tracks (subpleural linear opacity connecting pleura and peripheral nodule or mass), and cysts [8–10]. However, as these radiologic findings are similar to those of tuberculosis, lung cancer, and other pulmonary infections, delayed diagnosis, misdiagnosis, and missed diagnosis occur frequently. Many patients with pleuropulmonary paragonimiasis face a considerable burden of long-term hospitalization, as well as unnecessary procedures and treatments, before receiving the correct diagnosis [11–13]. Recently, several clinical cases of delayed diagnosis or misdiagnosis of pleuropulmonary paragonimiasis with atypical features were shown. Hwang et al. reported [14] that they observed clinical cases of pleuropulmonary paragonimiasis masquerading as pleural tuberculosis. Moreover, Shu et al. [15] found that eight pleuropulmonary paragonimiasis patients experienced delayed diagnosis due to symptoms shared with lung masses, six were misdiagnosed as having pneumonia, and two were misdiagnosed as having pulmonary tuberculosis.

Although radiologic findings of pleuropulmonary paragonimiasis disease easily overlap with other diseases, compared with thoracentesis or biopsy, computed tomography (CT) still plays significant role in diagnosing pleuropulmonary paragonimiasis. The majority of papers focus on retrospectively analyzing the data from patients diagnosed with pleuropulmonary paragonimiasis in several decades. However, there are no reports concerning a delayed diagnosis of pleuropulmonary paragonimiasis on CT scans. Given the above background, we conducted this multi-center study to identify the CT findings that are responsible for a delayed diagnosis of pleuropulmonary paragonimiasis.

## 2. Materials and Methods

The Institutional Review Board (IRB 2012-05-024) of our study centers approved this study. This study was conducted under a retrospective design, using an informed questionnaire survey. Therefore, our clinical series of patients did not need to submit a written informed consent form. An electronic survey with several topics (personal information, clinical findings, method of diagnosis, type of CT equipment, CT findings, and the initial impression on CT) was carried out at each hospital in 103 patients. Our clinical series of patients consisted of 58 men and 45 women, whose mean age was $46.1 \pm 14.6$ years (range, 12–88 years). They were diagnosed with pleuropulmonary paragonimiasis during a period ranging from 2003 to 2008. Data collection and statistical analysis were performed at a single center.

Questionnaires included the following: (a) age and sex, (b) place of residence, (c) underlying lung diseases, and (d) a past history of eating raw or undercooked shellfish. We also examined whether our clinical series of patients had underlying lung diseases to identify the presence of tuberculosis, emphysema, lung malignancy, and interstitial lung disease, among others. Clinical laboratory findings included (a) white blood cell (WBC) counts, (b) differential counts, and (c) absolute eosinophil counts. Of the 103 patients in total, 97 were available for an analysis of the clinical laboratory data. Eosinophilia was defined as the absolute eosinophil counts of $\geq$500/mL.

Our clinical series of patients were subjected to CT scanning with a single detector row scanner (Siemens Medical Solutions, Forchheim, Germany) ($n = 17$), 4 (LightSpeed QX/I, GE Healthcare, Milwaukee, WI, USA) ($n = 27$), 16 ($n = 54$), or 64 multi-detector CT scanners ($n = 5$) (Definition 16 or Sensation 64; Siemens Medical Solutions, Forchheim, Germany). CT scanning parameters were as follows: a single detector row scanner (collimation, 5 mm; tube voltage, 100–120 kV; tube current, 180–200 mA; gantry rotation time, 1 s; and pitch, 1.5); 4 MDCT scanner (collimation, 5 mm; tube voltage, 120–140 kV; tube current, 160–200 mA; gantry rotation time, 0.8 s; and pitch, 1.5); 16 or 64 MDCT scanner (collimation, 0.5 mm; tube voltage, 120 kV; tube current, 200 mA; gantry rotation time, 0.5 s; and pitch, 1.1). The images were reviewed for the current analysis, and were then reconstructed using a standard reconstruction algorithm with a slice thickness of 5 mm contiguously. All images were reviewed on the picture archiving and communication system (PACS) workstations using axial images, with a window level of 50 Hounsfield units and a window width of

350 Hounsfield units. Both non-contrast- and contrast-enhanced CT scans were obtained from 85 patients. For an enhancement study, a 100 mL of contrast medium (Ultravist 370; Schering, Berlin, Germany; Visipaque, Germany; GE Healthcare, Milwaukee, WI, Omniscan, USA; GE Healthcare, Milwaukee, WI, USA) was administered intravenously.

The parenchymal findings on CT scans were recorded for each lesion: (a) location (five lobes), (b) distribution (central, peripheral, and both), (c) type of lesions (non-cavitary nodule, cavitary nodule, consolidation, mass, linear opacity, and others), (d) margin of lesions, (e) lesion multiplicity, (f) size, and (g) pattern of enhancement (no enhancement, homogenous enhancement, heterogeneous enhancement, including central low attenuation, and perilesional ground glass opacity). The longest diameters of lesions were measured. In addition, known as one of the typical findings of pleuropulmonary paragonimiasis, the presence or absence of the sign of worm migration tracks was evaluated in 92 patients. If there were any signs, the corresponding patients were subjected to a measurement of the length and diameter. We also evaluated the presence or absence of pleural lesions in 102 patients. Finally, we analyzed initial diagnoses made solely on CT scans.

A binomial test was performed to identify statistical differences in such variables as age, sex, a history of eating raw or undercooked shellfish, eosinophilia, location, distribution, type of lesions, margin, and the pattern of contrast-enhancement between patients with an initially correct diagnosis and those with an incorrect one. The diagnostic accuracy of CT scans in patients with pleuropulmonary paragonimiasis was calculated by comparing the rates of diagnosis between the CT scans and the histological and clinical diagnoses. A $p$-value of $<0.05$ was considered statistically significant. Statistical analysis was performed using R software version 3.6.3 (R Foundation for Statistical Computing, Vienna, Austria).

## 3. Results

The clinical characteristics of patients with pleuropulmonary paragonimiasis are summarized in Table 1. Only 14 patients (13.9%) had a history of eating raw or undercooked shellfish, but the remaining patients had no clear history. The diagnostic methods for pleuropulmonary paragonimiasis are summarized in Table 2. The enzyme-linked immunosorbent assay (ELISA) test was performed in 97 patients (95.1%), 99% of whom were positive for ELISA, being the highest rate of diagnosis, as compared with other tests (range, 5.0 to 43.2).

**Table 1.** Clinical characteristics of patients with pleuropulmonary paragonimiasis.

| Type of Questionnaire | Response Rates | Responses (*n*, %) |
|---|---|---|
| History of eating | 98.1% (101/103) | yes (*n* = 14, 13.9), no (*n* = 8, 7.9), unknown (*n* = 79, 78.2) |
| Leukocytosis * | 99.0% (102/103) | yes (*n* = 25, 24.5), no (*n* = 77, 75.5) |
| Eosinophilia ** | 94.2% (97/103) | yes (*n* = 84, 86.6), no (*n* = 13, 13.4) |
| Symptoms * | 100% (103/103) | blood-tinged sputum (*n* = 23, 23.3), hemoptysis (*n* = 18, 17.5), dyspnea (*n* = 23, 23.3), chest pain (*n* = 25, 24.3), cough (*n* = 15, 14.6), fever (*n* = 3, 2.9), sputum (*n* = 3, 2.9), abdominal pain (*n* = 2, 1.9), hemiparesis (*n* = 2, 1.9), no symptoms (*n* = 11, 10.7) |

* Leukocytosis > 10,800/μL, ** Eosinophilia > 500/μL, * More than one symptom may be listed.

**Table 2.** Diagnostic methods for patients with pleuropulmonary paragonimiasis.

| Diagnostic Tools | Rates of Compliance (%) | Positivity Rates (%) |
|---|---|---|
| Sputum examination (*n* = 39) | 38.2 | 15.4 |
| ELISA (*n* = 97) | 95.1 | 99.0 |
| Bronchoscopy & BAL* (*n* = 40) | 39.2 | 5.0 |
| Tissue biopsy (*n* = 44) | 43.1 | 43.2 |

* BAL: bronchoalveolar lavage.

The CT findings of pleuropulmonary paragonimiasis are summarized in Table 3. Of the 103 patients, 91.3% (94/103) were available for an analysis of the location and

distribution of the lesions. The main pulmonary lesions were located in the upper lobe in 57.4% (54/94) (Figure 1a,b), in the lower lobe in 35.1% (33/94), and in the right middle lobe in 7.5% (7/94) (Figure 2a,b), with a peripheral distribution in 77.7% (73/94), a central distribution in 13.8% (8/94), and both in 8.5% (3/94). Nodular opacities on CT scans were the most common findings (53/94, 56.4%). By analyzing the patterns of nodules, non-cavity nodule and cavitary nodules were reported to be 59.6% (31/53) and 40.4% (22/53), respectively. The sign of worm migration tracks, known as one of the typical findings of pleuropulmonary paragonimiasis, was only present in 18.1% (17/94) of patients. The mean length and diameter of worm tracks were 19.7 mm and 5.3 mm, respectively (Figure 3a,b) Although the initial CT showed no statistical significance between the group with correct diagnostics and the group with incorrect diagnostics, in the case of a correct diagnosis, the number migrating worm track was high (12/17, 70.6%), and in the group showing incorrect diagnosis, air-space consolidation (18/25, 72%), and mass (6/8, 75%) were high.

**Table 3.** CT findings of pulmonary parenchymal lesions in patients with pleuropulmonary paragonimiasis.

| Parenchymal Findings | *n* (%) | Initial CT Diagnosis | | *p* Value * |
|---|---|---|---|---|
| | | Correct (%) | Incorrect (%) | |
| Locations (*n* = 94) | | | | |
| upper lobes | 54 (57.4) | 26/54 (48.1) | 28/54(51.9) | 0.481 |
| lower lobes | 33 (35.1) | 15/33 (45.5) | 18/33 (54.5) | 0.454 |
| right middle lobes | 7 (7.5) | 4/7 (57.1) | 3/7 (42.9) | 0.571 |
| peripheral | 73 (77.7) | 36/73 (49.3) | 37/73 (50.7) | 0.493 |
| central | 13 (13.8) | 7/13 (53.8) | 6/13 (46.2) | 0.538 |
| both | 8 (8.5) | 2/8 (25) | 6/8 (75) | 0.250 |
| findings (*n* = 94) | | | | |
| nodules | 53 (56.4) | 31/53 (58.5) | 22/53 (41.5) | 0.585 |
| non-cavitary | 31 (33.0) | 16/31 (51.6) | 15/31 (48.9) | 0.516 |
| cavitary | 22 (23.4) | 15/22 (68.2) | 7/22 (31.8) | 0.682 |
| air-space consolidation | 25 (26.6) | 7/25 (28) | 18/25 (72) | 0.280 |
| mass | 8 (8.5) | 2/8 (25) | 6/8 (75) | 0.250 |
| linear density | 5 (5.3) | 3/5 (60) | 2/5 (40) | 0.600 |
| others | 3 (3.2) | | | |
| migrating worm track (*n* = 94) | | | | |
| presence | 17 (18.1) | 12/17 (70.6) | 5/17 (29.4) | 0.706 |

Note: Data in parentheses are percentages. * Analyzed by binomial test (H$_0$: The correct proportion is 0.5).

The pleural findings are summarized in Table 4. Of the 103 individuals, 99% (102/103) were available for an analysis of the pleural abnormalities. In our study, the prevalence of pleural abnormalities was 57.8% (59/102). Common pleural findings include pleural effusion in 67.8% (40/102), and diffuse pleural thickening in 35.6% (21/102). However, the incidence of hydropneumothorax (15.3%, 9/102) or pneumothorax (6.8%, 4/102) was relatively lower. Of these pleural abnormalities, however, only hydropneumothorax (*p* = 0.04) had a significant correlation with a correct diagnosis of pleuropulmonary paragonimiasis.

In 44.7% (46/103) of our clinical series of patients, a correct diagnosis of pleuropulmonary paragonimiasis was initially made on CT scans. Tentative diagnoses were also made on CT scans, and these included pneumonia in 17.5% (18/103), tuberculosis in 13.6% (14/103), and lung cancer in 6.8% (7/103) (Figure 4a–d) (Table 5).

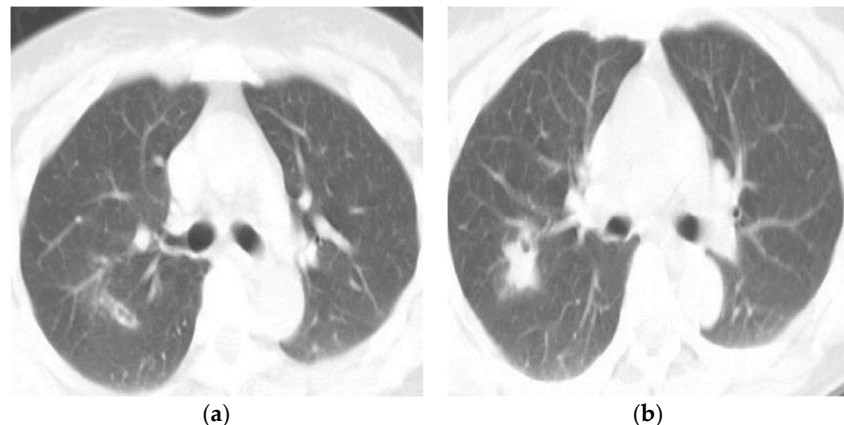

**Figure 1.** A 41-year-old woman was admitted to an emergency room with a chief complaint of persistent cough, presenting with eosinophile counts of 14.4 %. Chest CT scans of the lung (**a**,**b**) show a focal consolidation with a migration worm track in the right upper lobe. The patient was initially diagnosed with pleuropulmonary paragonimiasis on CT scans, whose findings were positive in the enzyme-linked immunosorbent assay (ELISA).

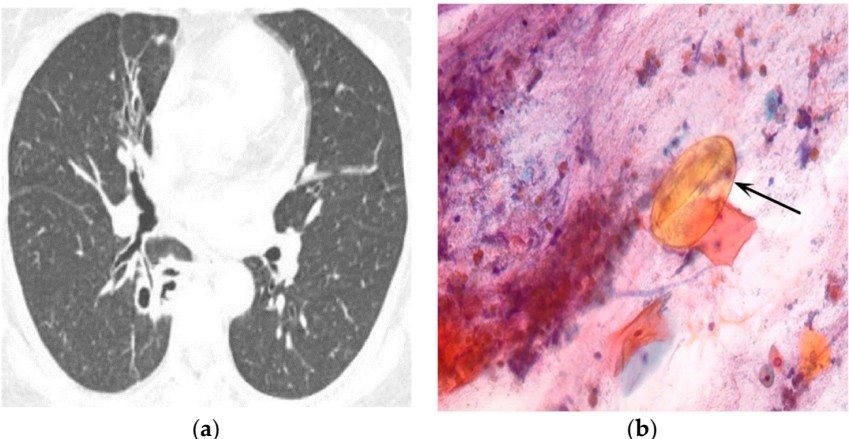

**Figure 2.** A 56-year-old woman presented with a 1-month-history of hemoptysis once a day. Chest CT scans of the lung (**a**) show multiple cavities with a varying size, which is accompanied by wall thickening in the superior segment of the right lower lobe and the focal bronchiectasis in the right middle lobe. On sputum examinations (**b**), there are round, yellow eggs (black arrow).

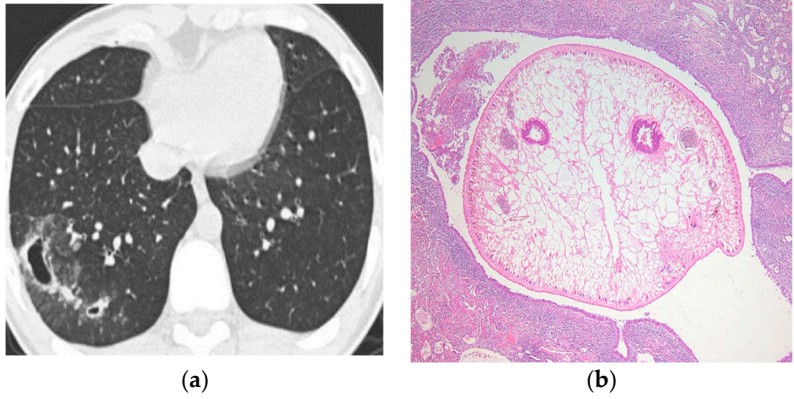

**Figure 3.** A 21-year-old man suffered from a 1-year-history of intermittent bleed-tingled sputum. Chest CT scans of the lung (**a**) show a large cavity of around ground glass opacity in the right lower lobe. (**b**) The patient underwent open lung biopsy, by which manner a worm was detected on a pathologic specimen.

**Table 4.** CT findings of pleural lesions in patients with pleuropulmonary paragonimiasis.

| Pleural Findings (*n* = 102) | Numbers (%) | Initial CT Diagnosis | | *p* Value |
|---|---|---|---|---|
| | | Correct | Incorrect | |
| absence | 43 (42.2) | 17/43 (39.5) | 26/43 (60.5) | 0.395 |
| presence | 59 (57.8) | 29/59 (49.2) | 30/59 (50.8) | 0.492 |
| pleural effusion | 40 (67.8) | 16/40 (40) | 24/40 (60) | 0.400 |
| diffuse pleural thickening | 21 (35.6) | 7/21 (33.3) | 14/21 (66.7) | 0.333 |
| hydropneumothorax | 9 (15.3) | 6/9 (66.7) | 3/9 (33.3) | 0.667 |
| focal pleural thickening | 8 (13.6) | 4/8 (50) | 4/8 (50) | 0.500 |
| pneumothorax | 4 (6.8) | 2/4 (50) | 2/4 (50) | 0.500 |

Note: Data in parentheses are percentages.

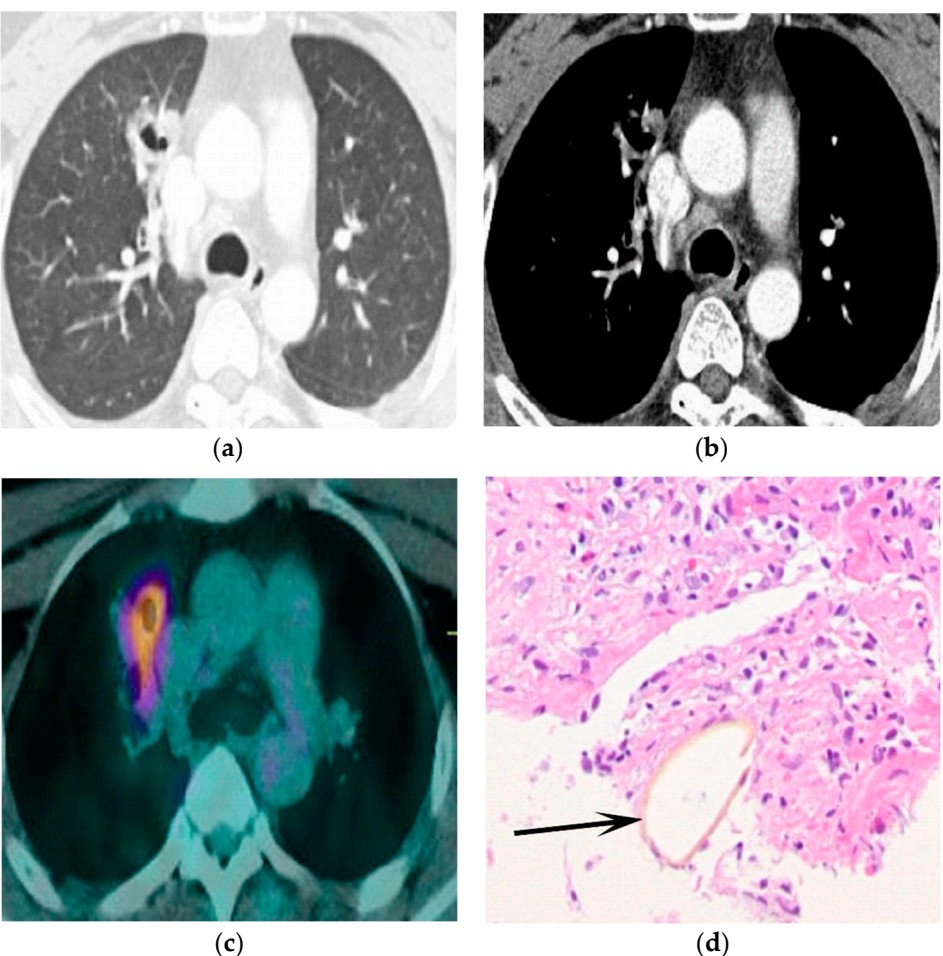

**Figure 4.** A 39-year-old man had a 6-month-history of intermittent hemoptysis, who was a heavy smoker. (**a**) Chest CT scans of the lung show a cavitary mass with a wall thickening in the anterior segment of the right upper lobe. (**b**) Contrast-enhanced chest CT scans (mediastinal setting) show a mild enhancement of cavity and multiple, mildly enhanced mediastinal lymph nodes in the right paratracheal region. (**c**) The PET/CT shows a hypermetabolism of a cavitary mass. The patient was initially diagnosed on CT and PET/CT scans, which is suggestive of lung malignancy. (**d**) The patient underwent a percutaneous needle biopsy, by which an egg (black arrow) was detected on a pathologic specimen.

**Table 5.** Initial diagnoses made on chest CT scans.

| Initial Diagnosis | Numbers |
|---|---|
| pleuropulmonary paragonimiasis | 46 (44.7) |
| pneumonia | 18 (17.5) |
| tuberculosis | 14 (13.6) |
| lung cancer | 7 (6.8) |
| pleural effusion | 7 (6.8) |
| solitary pulmonary nodule | 4 (3.9) |
| lung abscess | 4 (3.9) |
| eosinophilic pneumonia | 3 (2.9) |

Note: Data in parentheses are percentages.

## 4. Discussion

The clinical and CT findings of pleuropulmonary paragonimiasis depend on the number of parasites present and the stage of infection. Im et al. [16] reported that an air-space consolidation and nodules were noted on CT scans in 82% and 41% of patients, respectively. In addition, Kim et al. [8] reported that the most frequent CT features of pleuropulmonary paragonimiasis are seen in 87% of cases, and these include subpleural or sub-fissural nodules, containing a necrotic low-attenuation area, with an ill-defined margin of around 2 cm in diameter. In addition, Im et al. [16] reported that there was a positive sign of worm migration tracks on CT scans in 41% of total patients. Moreover, according to Kim et al. [8], 48% of the patients were positive for worm migration tracks on CT scans. In our study, however, 18.5% of the patients responded that they had a worm migration track on CT scans. This suggests that one of the CT findings initially led to an incorrect diagnosis, and this might contribute to lowering the incidence of worm migration tracks in our clinical series of patients. As compared with previous reports, our results showed that the diagnostic rate was lower (< 50 %) on CT scans in patients with pleuropulmonary paragonimiasis [8,16]. Lee et al. [17] reported that the cavitary change and bronchial dilatation in infected dogs with pleuropulmonary paragonimiasis appeared on day 30. These authors also noted that there was a persistent presence of subpleural ground-glass opacities and nodules either with or without cavitary changes until day 180 on micro-CT scans. After a cavitary change of the nodules, the sign of worm migration tracks was detected on all the serial CT scans. Clinically, worm migration into the thoracic cavity can occur 3–8 weeks after ingestion of metacercaria. In the current study, however, most of the patients (69.2%) were admitted within four weeks since the onset of symptoms. Therefore, the worm migration track could not be detected. This might have led to a delayed diagnosis of pleuropulmonary paragonimiasis on CT scans.

Typical CT findings of pleuropulmonary paragonimiasis were associated with subpleural nodules with an ill-defined margin, containing a low-attenuated necrotic area, since the pathogens' eggs occlude the small-sized vessels in the adjacent parenchyma [8,16,18]. Similar CT findings include active pulmonary tuberculosis and lung malignancy, both of which had central areas of low density on contrast-enhanced CT scans [19–23]. Overall, 61% of the patients enrolled in our study showed a central low attenuation on contrast-enhanced CT scans. This led to a misdiagnosis pleuropulmonary paragonimiasis. Our results showed that there was a central low attenuation on contrast-enhanced CT scans at a proportion of 53.8% (43/80). However, these results reached no statistical significance regarding the difference between a correct diagnosis and an incorrect one.

Cavity can have a variety of causes, including neoplasm, infection, granulomatous disease, and collagen vascular disease [19,24,25]. In addition, cavitation in lung malignancy has been reported to have been seen in up to 22% of total cases, and more commonly with squamous cell carcinoma [26–28]. According to Kim et al. [8], the nodule on CT scans in pleuropulmonary paragonimiasis appears as a cavity, often registering in up to 58% of frequency. In our study, 23.4% of the nodules had a cavitary form on CT scans. Following an analysis of the location of the lesion in our clinical series of patients, the upper lobe

was found to be common (57.4%). In 20.4% of patients with upper lobe predominance for whom an incorrect diagnosis was made, a misdiagnosis of tuberculosis or malignancy was initially made, although their CT scans were reviewed by board-certified specialists in thoracic radiology. Based on these results, it can be inferred that it may be difficult to make a differential diagnosis between differentiate tuberculosis and lung malignancy if the lesion involves the upper lobes with cavitation. Presumably, this might lead to a misdiagnosis of pleuropulmonary paragonimiasis in endemic areas.

Previous reports have shown that a pleural lesion of paragonimiasis is also one of the common clinical manifestations [8,9,16,29]. The prevalence of pleural effusion in patients with pleuropulmonary paragonimiasis varies, ranging from 2.9 % to 54% [16]. By contrast, Kim et al. [8] reported that focal pleural thickening and effusion or hydropneumothorax were seen in 87% and 29% of total cases, respectively. Im et al. [16] reported that unexplained bilateral pleural effusion or hydropneumothorax are suggestive of pleuropulmonary paragonimiasis in endemic areas. The causes of hydropneumothorax include iatrogenic factors (the influx of air during pleural effusion aspiration), the presence of gas-forming organisms, and trauma. In our study, the incidence of hydropneumothorax was 15%, and it was a key diagnostic clue for pleuropulmonary paragonimiasis.

In making a diagnosis of pleuropulmonary paragonimiasis, a past history of eating raw or undercooked shellfish would be a key clue. Patients should be suspected of having pleuropulmonary paragonimiasis if they are residing in, or have traveled to, endemic areas. In the current study, however, it was uncertain whether most of the patients had a past history of eating undercooked shellfish. Moreover, a past history of eating undercooked shellfish was not helpful for making a diagnosis of pleuropulmonary paragonimiasis. According to Cho et al. [3] and Meehan [30] et al., there are others sources of pleuropulmonary paragonimiasis, and these include Kejang (drunken crab), the juice from freshwater crayfish, eating raw meat or raw intermediate hosts, and raw crabmeat which is imported in a pickled form. Therefore, clinicians should perform a variety of questionnaire studies, depending on where patients have a past history of eating raw or undercooked shellfish in countries where there are diverse eating habits.

Clinicians sometimes consider pleuropulmonary paragonimiasis in making a differential diagnosis of patients with eosinophilic pleuropulmonary disease [29,31]. In the current study, 84.5% of total patients had a peripheral eosinophilia. Despite the definite presence of parenchymal abnormalities, however, around 15% of total patients showed normal eosinophil counts in the peripheral blood. All of these patients had a delayed diagnosis of pleuropulmonary paragonimiasis. If there are any findings that are suggestive of eosinophilic pneumonia, but no consistently specific findings in the organisms and/or eggs on biopsy, serologic tests would provide a key diagnostic clue. It is well known that a diagnosis of pleuropulmonary paragonimiasis is established based on positive ELISA results. ELISA is commonly used for an immunodiagnostic of pleuropulmonary paragonimiasis, and it is the most sensitive method that produces a sensitivity of 80% and a specificity of 97% [32]. Our results showed that the positive rate was very low on sputum examination and bronchial-washing cytology. It was also shown that this was not exceeded on lung biopsy. However, ELISA showed a detection rate of almost 100%. This strongly indicates that ELISA should be performed for patients who are suspected of having pleuropulmonary paragonimiasis from endemic areas.

The limitations of the current study are as follows: (1) The current questionnaire study was conducted under a retrospective design. We therefore failed to evaluate where there could be an inter-observer variation, including the radiologist's experience in the assessment of CT findings. We could speculate, however, that an inter-observer variation in the assessment of CT findings would be negligible, since our clinical series of patients were studies by board-certified specialists in chest radiology who had experience in making a diagnosis of pleuropulmonary paragonimiasis, and who had long lived in endemic areas of pleuropulmonary paragonimiasis. (2) A diagnosis of pleuropulmonary paragonimiasis should be made based on the appropriate clinical information and laboratory data. In the

current study, however, we analyzed only an initial diagnosis made on CT scans, without considering the clinical and laboratory findings of patients. Our results showed that the initial diagnostic rate for pleuropulmonary paragonimiasis was relatively lower on CT scans. However, this could be considered unacceptable by other radiologists. Henceforth, further questionnaire studies are therefore warranted to overcome our limitations in estimating the exact diagnostic rate for pleuropulmonary paragonimiasis on CT scans.

## 5. Conclusions

In conclusion, a misdiagnosis of pleuropulmonary paragonimiasis is often made if the lesions contain a pneumonic consolidation or lung mass, as well as if there are lesions in the upper lobes. It would be mandatory to combine the clinical and laboratory manifestations with chest CT characteristics, which would be essential for making a diagnosis of pleuropulmonary paragonimiasis in patients without specific signs of the disease on CT scans.

**Author Contributions:** Conceptualization, K.K.L.; G.Y.J.; methodology, G.Y.J.; K.S.K.; software, K.S.K.; validation, G.Y.J.; K.S.K.; formal analysis, K.S.K.; investigation, K.K.L.; G.Y.J.; resources, K.K.L.; G.Y.J.; data curation, G.Y.J.; K.S.K.; writing—original draft preparation, K.K.L.; G.Y.J.; writing—review and editing, K.K.L.; G.Y.J.; visualization, G.Y.J.; supervision, G.Y.J. All authors have read and agreed to the published version of the manuscript.

**Funding:** This research received no external funding.

**Institutional Review Board Statement:** This was a retrospective study approved by Research Institute of Clinical Medicine, Biomedical Research Institute of Jeonbuk National University Hospital, Jeonbuk National University Medical School, Institute of Medical Science (IRB 2012-05-024).

**Informed Consent Statement:** Patient consent was waived due to being retrospective study concerning past chest CT.

**Data Availability Statement:** Data are available in a publicly accessible repository.

**Acknowledgments:** Thank you Yun Hyeon Kim, Hyun Ju Seon, Mi Sook Lee, and Seong Hoon Park, for their cooperation in this survey. Thank you as well to Sung-Kwan Kim, who worked at Jeonbuk National University Hospital, for helping me organize data, organize statistics, and write our paper.

**Conflicts of Interest:** The authors declare no conflict of interest.

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
