# Peer review of "What Findings on Chest CTs Can Delay Diagnosis of Pleuropulmonary Paragonimiasis?"

_tomography, doi:10.3390/tomography8030122_

Round 1

Reviewer 1 Report

The paper is aimed at the identification of findings in the chest CT findings of paragonimiasis that delayed the diagnosis of this disease. The topic is interesting and of importance in Asia, Africa, and South America. The experiments are clearly described as well as correctly designed and performed. Thus, presented conclusions are sound. There just a few minor issues that should be clarified before the paper will be suitable for publication.
1.    How experienced were the radiologists who analysed CT scans?
2.    The image analyses were performed manually, or some image processing/analysis algorithms were applied? If not, development of such algorithms is suitable?
3.    Is it possible based of the presented results to develop some diagnostic protocol that would improve the sensitivity of the diagnosis of pleuropulmonary paragonimiasis?

Reviewer 2 Report

Thank you for the opportunity to review this manuscript on diagnosis of pleuropulmonary paragonimiasis. I have provided some comments for consideration below.

32: Italicize “Paragonimus”

43: Italicize “Paragonimus westermani”

44: Italicize “P. westermani”

58-60: Numbers < 10 should be spelled out in full

129: 101/103 (98.1%) is already shown in Table 1 – do not repeat here

Table 1: Column 1 should not be centered; do not repeat n/% to avoid redundancy – that it, change column 3 header to “Responses (n, %)” and report rows as, for example “yes (14, 13.9); no (8, 7.9); unknown (79, 78.2)”

Table 2: This is misleading, as percentages do not sum to 100%. Therefore, you need to indicate that patients may have received more than one diagnostic method. In other words, diagnosis appears to be a multiple response variable. Is this correct?

Table 2: Do not center column 1

140-152: You are repeating data that is already in the table.

149: -152: You use the word “correlation” but did not perform a correlation; use the correct statistical language here

Table 3: Do not center column 1. “Numbers (%)” should be “N (%).” Also, lesion location appears to be a multiple response variable, making it difficult to understand the table.

Table 3 – methods: I do not think you have analyzed this properly. You are comparing correct vs. incorrect diagnosis based on several findings. However, since there are only two groups, being tested, once the proportion of correct is known, the proportion of incorrect is fixed – so the two proportions are not functionally independent. What you really want to test is whether the proportion differs from 50:50 (equivalent to a 1:1), which is the null of no difference. Examining “migrating worm track” as an example, you can test this using R with the script <binom.test(x=c(12,5),p=0.5,alternative="two.sided")>. This leads to an estimated proportion of 0.7059, with a 95% CI of (0.4404,0.8969), and an exact p-value=0.1435. In this case, even though “70.6%” seems much greater than “29.4%”, the sample size does not render a significant difference, and asymptotic p-values are inappropriate, anyway, so the exact binomial is the best choice. You may consider a slightly different approach – can you model correct vs. incorrect diagnosis as the outcome, and your findings and patient characteristics as your predictors. In this way, you can examine the odds of making correct or incorrect clinical diagnosis based on known characteristics of patients, lesion location, worm tracks, etc.

Additionally, if parenchymal findings are multiple-response variables, why are you testing them independently? For example, are there any cases where lesions in a single patient that appear both in the upper and lower lobes showed different CT diagnosis results?

Table 4: Do not center column 1; See comment above – I believe that your statistical analysis is incorrect.

Table 5: Do not center column 1.

278-280: ELISA tests are standard, whereas sputum-egg detection and other methods are quite well known to be less sensitive to diagnosis of paragonimiasis. Given this is an old data set, I am curious if diagnostic methodology has changed over time? In other words, how informative is a retrospective data analysis of data from 15-20 years ago compared to standard care today? Additionally, CT scanning technology has changed demonstrably as well in that time, so the relevance of this data set compared to current methodology is questionable. Please address these issues.

Round 2

Reviewer 2 Report

You have addressed the original concerns and provided adequate responses.